environmental chemistry

phosphorus-doped boron nitride, doping modification, heavy metal, adsorption, selectivity

**Author for correspondence:**
Rundong Li
e-mail: rdlee@163.com

This article has been edited by the Royal Society of Chemistry, including the commissioning, peer review process and editorial aspects up to the point of acceptance.

†Present address: No. 37 Daoyi South Avenue, Daoyi Development District, Shenyang China.

# Preparation of phosphorus-doped boron nitride and its adsorption of heavy metals from flue gas

Yanlong Li[1,2], Hongxi Li[2], Rundong Li[1,2,†], Xin Su[2] and Shengqiang Shen[1]

[1]Key Laboratory of Ocean Energy Utilization and Energy Conservation of Ministry of Education, School of Energy and Power Engineering, Dalian University of Technology, Dalian 116024, People's Republic of China
[2]Key Laboratory of Clean Energy, College of Energy and Environment, Shenyang Aerospace University, Shenyang 110136, People's Republic of China

(iD) YL, 0000-0002-9857-6563

Boron nitride, also known as white graphene, has attracted extensive attention in the fields of adsorption, catalysis and hydrogen storage due to its excellent chemical properties. In this study, a phosphorus-doped boron nitride (P-BN) material was successfully prepared using red phosphorus as a dopant for the preparation of porous boron nitride precursors. The phosphorus content in the P-BN was adjusted based on the addition rate of phosphorus. The specific surface area of P-BN first increased and then decreased with increasing addition rate of phosphorus. The maximum specific surface area was $837.8 \, \text{m}^2 \, \text{g}^{-1}$ when the phosphorus addition rate was 0.50. The P-BN prepared in the experiments was used as an adsorbent, and its adsorption capacity for heavy metals from flue gas was investigated. In particular, P-BN presented a stronger adsorption selectivity for zinc compared with other heavy metals, and its adsorption capacity for zinc was 5–38 times higher than for other heavy metals. The maximum adsorption capacity of P-BN for zinc and copper in a single heavy metal atmosphere was 69.45 and $53.80 \, \text{mg} \, \text{g}^{-1}$, respectively.

## 1. Introduction

Boron nitride is a novel porous material which has attracted the attention of many researchers [1–5]. It has been studied for hydrogen storage and adsorption applications due to its high specific surface area and chemical properties [6–9]. For example, Wang used boron nitride for hydrogen storage experiments and found that the materials exhibit high and reversible $H_2$ absorption

rate of 1.6 to 2.3 wt% at 77 K under a relatively low pressure [10]. The research and use of boron nitride have demonstrated its good adsorption effects on antibiotics and chemical dyes in water [11–14]. Such as the boron nitride nanomaterial prepared by Singla *et al.* adsorbed bright green and methyl orange and explained the feasibility of the interaction between the dye molecules and the adsorbed nanomaterial by density functional theory [15]; Song *et al.* [16] studied the adsorption of the boron nitride bundles on antibiotics and found that it has higher adsorption capacity and superior removal percentage than the carbonaceous adsorbent, and his team has demonstrated through repeated experiments that boron nitride bundles have better reusability. In addition, doping activation was becoming more and more popular in the field of materials. Such as that Shukla *et al.* [17] introduced carboxylated doping on the membrane material to increase the hydrophilicity of the membrane, thereby significantly improving the water flux and dye removal of the material, and Ali *et al.* [18] used silver-doped graphene oxide to enhance the biofouling resistance of the membrane material without affecting the selectivity of the membrane material. So the density functional theory research of boron nitride doping had gradually become the research focus [19,20]. Wang & Zhang [21] explored the adsorption of CO by silicon-doped boron nitride through the density functional theory. The results showed that the adsorption behaviour after doping changed from weak physical adsorption to strong chemical adsorption, with a significant increase of adsorption capacity. Research has also shown that the adsorption characteristics of different substances by boron nitride are different. Moreover, when other substances are introduced by doping, the average adsorption energy is reduced, or the binding ability of the adsorbent is improved [22,23], thereby improving its adsorption capacity.

Researchers have also observed that boron nitride presents good adsorption of heavy metal ions [24–26]. Wang *et al.* studied the adsorption of heavy metal ions in aqueous solutions. Their study showed that boron nitride could achieve high-adsorption equilibrium for $Pb^{2+}$ and $Cu^{2+}$ in a short period [27], and its adsorption capacity for these metals was 115.07 and 92.85 mg g$^{-1}$, respectively. In recent years, researchers have focused on the use of doped boron nitride for hydrogen adsorption and liquid phase contaminant removal [28,29]. However, there is a lack of research on the characteristics of heavy metal adsorption from flue gas by boron nitride. Due to the excellent thermal and chemical stability of boron nitride, its adsorption capacity for the removal of heavy metals from flue gas should be investigated.

The purpose of this research is to prepare a P-BN material with high specific surface area and a variety of pore structures by using the precursor method. Phosphorus was used as a dopant, and boric acid and melamine were used as boron and nitrogen sources. The obtained P-BN was used to absorb heavy metals from flue gas, and its absorption capacity in the field of flue gas purification was investigated.

# 2. Experiment

## 2.1. Preparation of phosphorus-doped boron nitride

For the experiments, 0.1–1 mol of phosphorus was added to a 1000 ml solution of melamine and boric acid mixed in a 1 : 2 molar ratio. The uniform mixture solution was placed in a constant temperature shaker at 80°C for 6 h to give a pink transparent solution. The pink precursor was obtained in a natural cooling mode. The precursor was suction filtered, washed, placed in a freeze dryer for 48 h and then placed in a tube furnace at 1150°C for pyrolysis experiments. In the experiment, $N_2$ was used as a protective atmosphere, at a flow rate of 100 ml min$^{-1}$. After cooling to room temperature, a fibrous P-BN was obtained.

## 2.2. Adsorption experiment of heavy metals from flue gas

The absorption experiments for heavy metals from flue gas using P-BN were performed in an independently designed experimental platform. A sufficient amount of heavy metal chlorides, used as the source of heavy metals, was placed in the constant temperature zone of the tube furnace, and the P-BN was placed in the heavy metal adsorption zone to absorb the heavy metals from the flue gas. In the course of the experiment, the basis weight of each heavy metal chloride was 5 g, and the basis weight of the adsorbent was 1.5 g. The tube furnace was heated to 950°C to ensure the occurrence of heavy metal fumes. The adsorbed P-BN was subjected to digestion experiments, using microwave-assisted acid digestion (HF-HNO$_3$-HClO$_4$). Then, the concentration of heavy metals in the digestion solution was analysed by ICP.

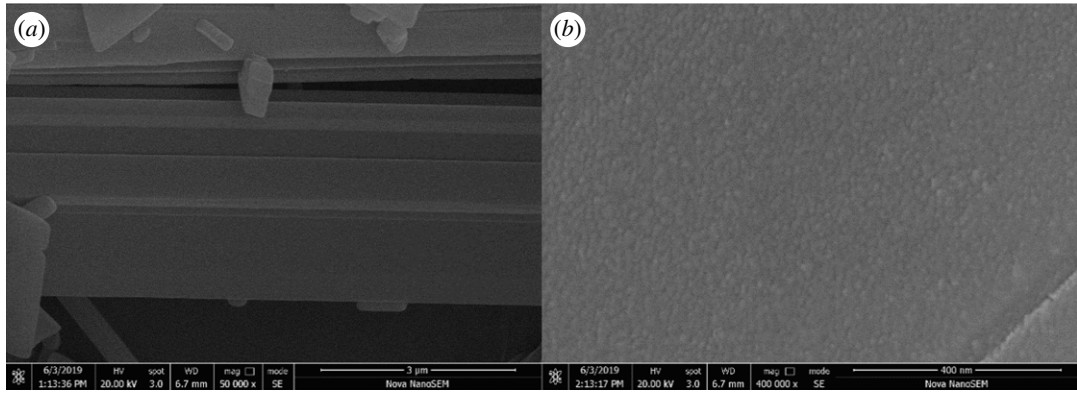

**Figure 1.** SEM image of boron-doped boron nitride (prepared using 0.5 mol of phosphorus): (*a*) layered structure and (*b*) rough surface topography.

## 2.3. Materials characterization

The morphology of P-BN was analysed by SEM. The pore structure and specific surface area were analysed by identifying the nitrogen adsorption isotherm using BET. The phosphorus doping amount and the obtained adsorption capacity for heavy metals at different doping ratios were detected by ICP.

# 3. Results and discussion

P-BN was successfully prepared by the precursor method using melamine and boric acid as basic materials and phosphorus as a dopant. The reactions that probably occurred during the experiment are described in equations (3.1) and (3.2).

$$2H_3BO_3 + C_3N_6H_66 + X\,P \rightarrow 2H_3BO_3 \cdot C_3N_6H_66 \cdot X\,P \tag{3.1}$$

and

$$2H_3BO_3 \cdot C_3N_6H_66 \cdot X\,P \xrightarrow{\Delta} 2P_xBN + 3H_2O + 2CO + 3NH_3 + N_2. \tag{3.2}$$

A SEM image of doped boron nitride (prepared with 0.5 mol of phosphorus) is shown in figure 1. The apparent lamellar structure of the boron nitride fibre is shown in figure 1*a*. It can be clearly seen from the figure that this layered structure can be more clearly reflected from the cross-sectional morphology of the P-BN surface. The rough surface structure of P-BN is the main factor leading to its high specific surface area, as shown in figure 1*b*. Such a rough surface structure provides richer adsorption sites and storage space for P introduced by doping, which can contribute to improve its absorption capacity [30]. P-BN was analysed by SEM-EDS to analyse the phosphorus doping effect. The SEM-EDS analysis revealed that there was a certain amount of phosphorus on the surface of P-BN, which preliminarily proved that the doping of phosphorus was successful.

The doping content of phosphorus was expressed by the doping ratio $\eta$. The phosphorus content of P-BN according to different doping ratios, which were obtained by ICP, is shown in table 1. As the doping ratio increased, the amount of phosphorus in the P-BN also increased. When the doping ratio reached 0.50, the amplitude of the lift tended to be small, and the phosphorus content at the doping ratio of 1 was only approximately 5 mg g$^{-1}$ higher than that at a doping ratio of 0.50. When the molar doping ratio exceeded 0.75, there was a significant P-precipitation phenomenon in the precursor solution. Therefore, based on the principle of green experiment, the optimal phosphorus effect was obtained at a doping ratio of 0.50, and the respective P-BN phosphorus content was 40.73 mg g$^{-1}$. The detection and analysis of ICP demonstrated the successful doping of phosphorus, which proved the feasibility of this doping preparation method.

The BET specific surface area of the P-BN samples at different doping ratios $\eta$ is shown in figure 2. The fluctuations seen in figure 2 indicate that the specific surface area showed a tendency of increasing first and decreasing afterwards according to the increasing doping of phosphorus. According to the data variation shown in the figure, the variation tendency of the specific surface area was the same when the calcination time was 4, 5 and 6 h. The specific surface area of the obtained P-BN was higher at a

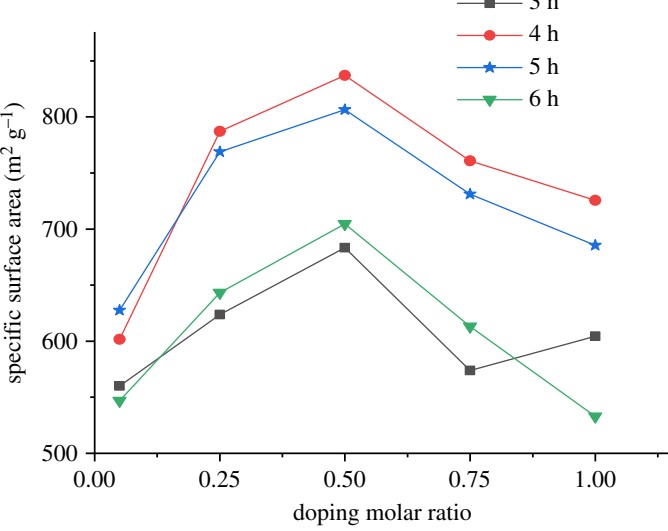

**Figure 2.** BET specific surface area of P-BN at different doping ratios.

**Table 1.** Content of P in P-BN upon different doping ratios.

| doping ratio $\eta$ (mol) | 0.1 | 0.25 | 0.50 | 0.75 | 1 |
|---|---|---|---|---|---|
| P content (mg g$^{-1}$) | 10.12 | 30.80 | 40.73 | 41.31 | 45.98 |

calcination time of 4 h. Therefore, 4 h was considered the optimum calcination time for the preparation of P-BN. Moreover, the P-BN obtained in each calcination stage exhibited the largest specific surface area at a doping ratio of 0.50. Therefore, it was concluded that the optimum doping ratio of the obtained P-BN was 0.50. In summary, the best preparation process of P-BN was obtained at a calcination time of 4 h and doping ratio of 0.50, at which the specific surface area was 837.08 m$^2$ g$^{-1}$.

The specific surface area and pore size distribution of all pyrolysis products (P-BN at different doping ratios) were investigated by measuring N$_2$ adsorption and desorption isotherms, and the results are shown in figure 3. The adsorption–desorption isotherm of P-BN obtained at a doping ratio $\eta$ of 0.50 was regarded as a type IV isotherm based on the International Union of Pure and Applied Chemistry (IUPAC) nomenclature classification [31,32], which exhibited a more obvious H4-type hysteresis loop. The image shows that the adsorption of N$_2$ presented a relatively rapid growth tendency in relatively low pressure. This behaviour indicates the presence of a microporous structure in the P-BN material. Moreover, the H4-type hysteresis curve is particularly obvious in the adsorption–desorption isotherm, and such characteristic in the adsorption–desorption isotherm indicates that a certain mesoporous structure exists in the P-BN [33]. As the relative pressure continued to increase up to 1, the adsorption of N$_2$ showed an increasing tendency again. The increased adsorption performance indicates that there were many macroporous structures in the P-BN material. The observed macroporous structure could have been caused by the adsorption sites provided by the P-doping, which was consistent with the more pronounced rough surface structure in figure 1$b$. Furthermore, the N$_2$ adsorption and desorption isotherms obtained at different $\eta$ doping ratios exhibited similar adsorption and desorption tendencies. The results indicate that P-BN prepared under different doping ratios $\eta$ presented the same pore structure distribution.

The specific surface area of boron nitride is not less than activated carbon, which is a typical flue gas adsorbent. And the B–N bonds of boron nitride have a local polarity that C–C bonds do not have and the presence of 28% of ionic bonds. After doping modification, boron nitride enhances the chemical adsorption capacity and further improves the adsorption capacity of boron nitride on the basis of the original physical adsorption capacity. Lvova & Ananina [34] discovered the existence of boron nitride vacancies through density functional theory. This phenomenon promotes the chemical activity of boron nitride. The defect regions formed due to the existence of vacancies can provide favourable conditions for the strong covalent bonding between the B and N atoms and the dissociated fragments of the adsorbed molecules. The existence of this vacancy mechanism provides boron nitride with space and

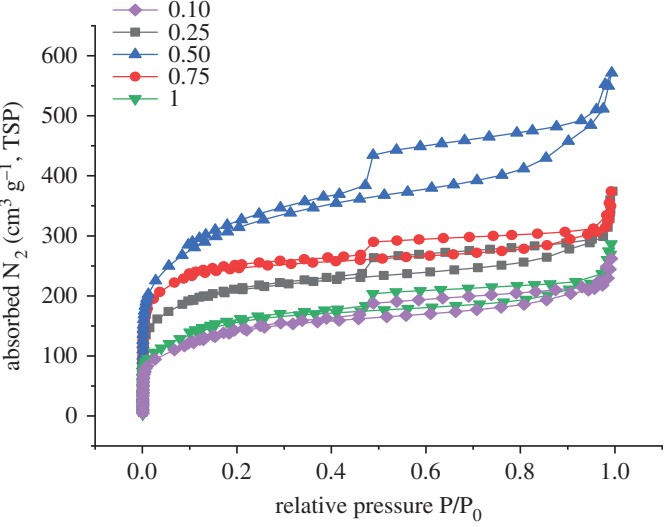

**Figure 3.** Adsorption and desorption isotherms for different doping ratios of P-BN.

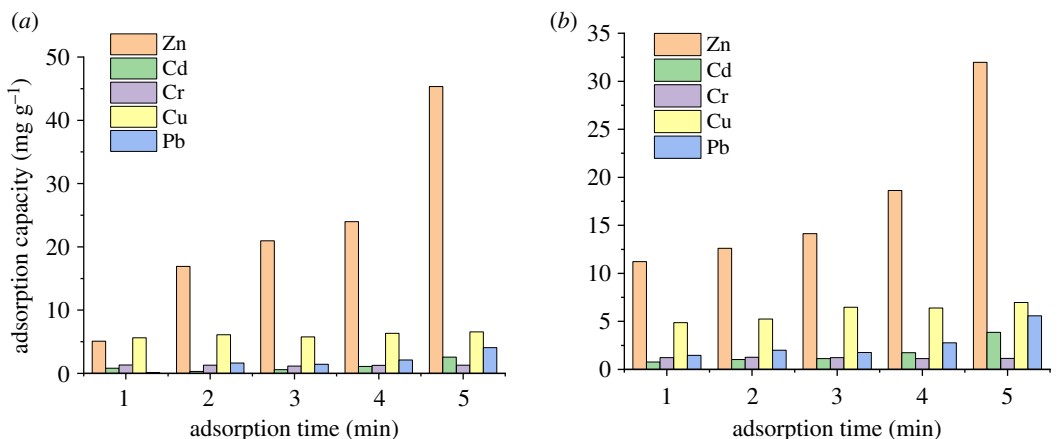

**Figure 4.** Adsorption capacity of P-BN for different heavy metals. (*a*) Adsorption capacity of P-BN for different heavy metals at 150℃. (*b*) Adsorption capacity of P-BN for different heavy metals at 200℃.

adsorption sites needed for adsorption. And boron nitride has high thermal stability and good chemical stability, so it is an effective material for adsorbing metals from flue gas. Heavy metals have the application value of precious metals in the field of resource utilization, and they have good practical research value for sustainable green development. Therefore, the adsorption capacity for different metals in flue gases, maximum adsorption capacity and adsorption selectivity of the P-BN adsorbent were studied. Adsorption experiments were conducted in a mixed flue gas environment prepared using various heavy metal salts and the P-BN obtained under optimal conditions.

The experimental data obtained by the digestion of adsorbed P-BN at different times and temperatures are shown in figure 4. Changes in the adsorption capacity of P-BN for heavy metal fumes according to adsorption time are shown for different adsorption temperatures in figure 4*a*,*b*. As the adsorption time increased, the adsorption capacity of P-BN for various heavy metals at different adsorption temperatures presented an increasing tendency. However, P-BN presented different adsorption selectivity for different heavy metals. The experimental data analysis showed that the adsorption characteristics of P-BN at different adsorption temperatures (50, 100, 250℃) had similar tendency, which were not listed here. The adsorption capacity of P-BN for zinc at different adsorption temperatures was far superior compared with those for other heavy metals. Meanwhile, the adsorption of copper was not as strong as that of zinc, but the adsorption capacity of copper was next to that of zinc at each adsorption temperature. The maximum adsorption capacity of P-BN for cadmium, chromium and lead was 4.55, 1.19 and 5.94 mg g$^{-1}$, respectively. The maximum adsorption capacity for copper and zinc was clearly higher than for other heavy metals, and was 9.53 and 45.33 mg g$^{-1}$, respectively, and the adsorption capacity for zinc

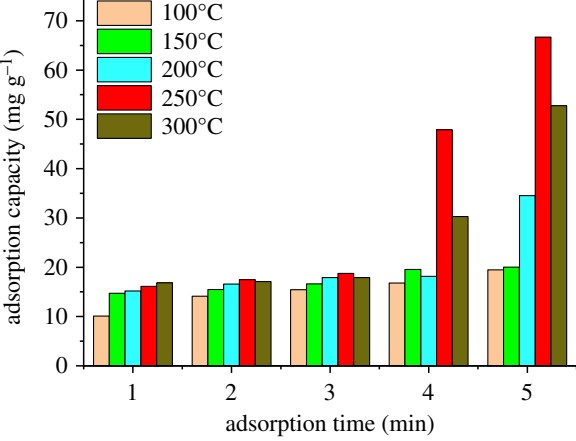

**Figure 5.** Adsorption of zinc by P-BN.

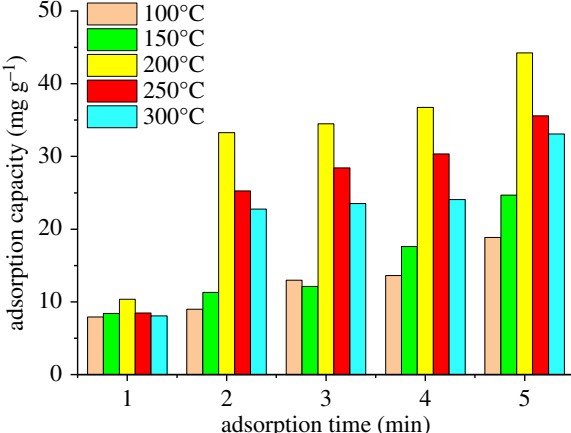

**Figure 6.** Adsorption of copper by P-BN.

was approximately five times higher than that for copper. Based on data comparison, it is clear that P-BN exhibits a certain adsorption capacity for heavy metals from flue gas, but it presents a different adsorption selectivity depending on the heavy metal. In particular, there was a stronger adsorption selectivity for zinc than for other heavy metals. Under the same experimental conditions, the adsorption capacity of P-BN for zinc was 5–38 times higher than that for other heavy metals.

To better study the adsorption capacity of P-BN for zinc and copper, we set the composition of the simulated flue gas as a single component. For that, a single-factor experimental method was used. The results are shown in figures 5 and 6. The adsorption capacity of P-BN for zinc at 250°C was the best at the same adsorption time. Such adsorption characteristics were especially clear as the adsorption time was extended. When the adsorption time reached 4 min, the adsorption capacity of P-BN for zinc suddenly increased to 47.89 mg g$^{-1}$. When the adsorption time increased to 5 min, the adsorbed amount of zinc reached 66.70 mg g$^{-1}$. The increased adsorption capacity of P-BN at 4 min may have occurred because the phosphorus doping provided better adsorption activity at that time, contributing to the improved adsorption capacity.

Under the same experimental conditions, the absorption tendency of copper was similar to that of zinc. However, the adsorption of copper was best at 200°C, and the adsorption capacity transition point occurred at 2 min. The maximum adsorption capacity of copper by P-BN was 44.23 mg g$^{-1}$. P-BN exhibited different adsorption transitions for different heavy metals. It was concluded that such different adsorption capacities were caused by the ability of phosphorus to adsorb different heavy metals. In this context, different transition times represented different energy requirements. The adsorption capacity transition indicated that the phosphorus in P-BN had good adsorption selectivity for copper, but as the time passed, P-BN exhibited a stronger adsorption capacity for zinc. According to the density functional theory, boron nitride presents different adsorption binding energies for

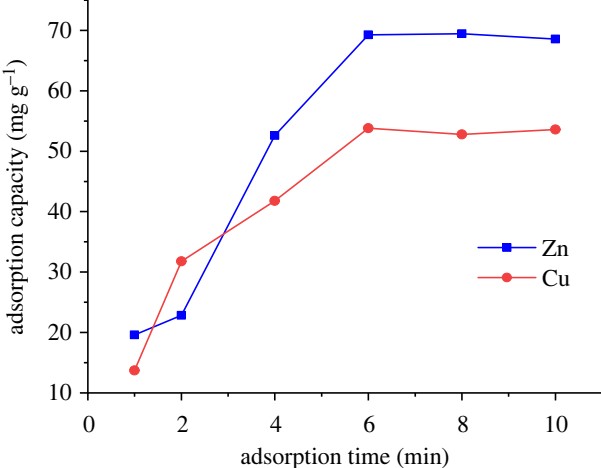

**Figure 7.** Saturation adsorption of zinc and copper by P-BN.

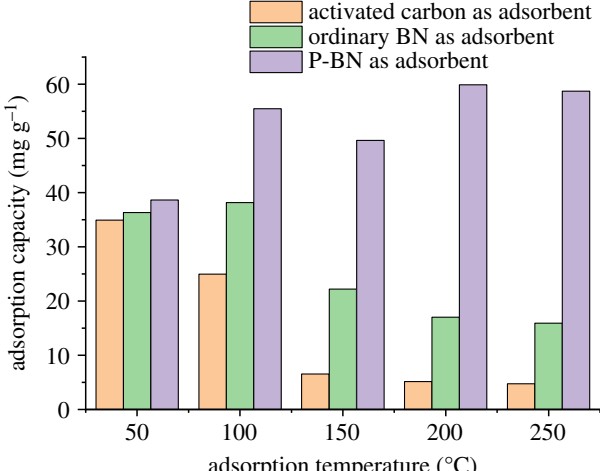

**Figure 8.** Total adsorption of heavy metals by different adsorbents.

different heavy metals, and platinum doping can reduce its average adsorption energy for hydrogen molecules. We observed a similar behaviour regarding the adsorption of heavy metals by P-BN in this experiment [22,35]. The reasons for the different adsorption capacities of different heavy metals by P-BN could be explained from the side.

The experimental data obtained by further exploring the adsorption capacity of zinc and copper by P-BN at optimum adsorption temperature are shown in figure 7. When the adsorption time increased from 1 to 2 min, the amount of adsorbed copper increased significantly. In this period, the adsorption capacity of P-BN for copper exceeded that for zinc. As the adsorption time increased to 4 min, the adsorption of zinc increased rapidly, while that of copper was slow. This adsorption characteristic was consistent with previous results, thus indicating that the phosphorus in the P-BN provided adsorption activity, and different adsorption selectivity existed for different heavy metals due to the adsorption energy required for adsorption targets. Moreover, these results were consistent with previous studies on density functional theory, which report that different adsorption energies are required by different heavy metals [35]. When the adsorption time increased again, the adsorbed amount gradually decreased after 6 min, then it reached adsorption saturation. The maximum amounts of adsorbed zinc and copper were 69.45 and 53.80 mg g$^{-1}$, respectively. In previous experiments, the maximum adsorption capacity of zinc by boron nitride prepared by the precursor method was 47.38 mg g$^{-1}$ during the adsorption of heavy metals from flue gas. The maximum adsorption capacity of P-BN was more than 20 mg g$^{-1}$ higher than that of ordinary boron nitride [36]. The total amount of adsorbed heavy metals by activated carbon, ordinary boron nitride and P-BN are shown in figure 8. It can be clearly seen that P-BN has good thermal stability and maintains good adsorption capacity during the

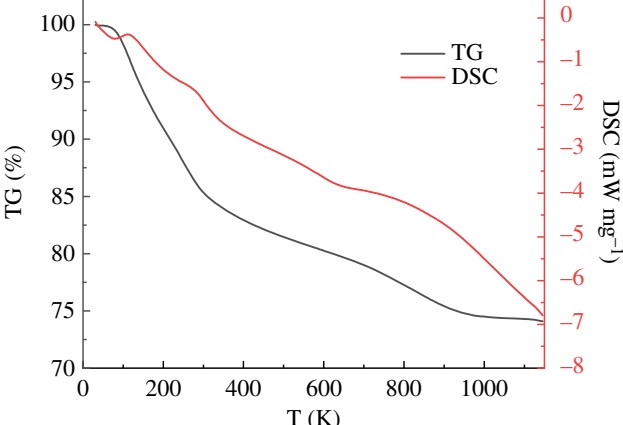

**Figure 9.** TG-DSC curve of P-BN after zinc adsorption (15°C/min, N$_2$ atmosphere).

temperature rise. When the temperature reaches 250°C, activated carbon loses its adsorption activity. However, at that temperature, the adsorption capacity of P-BN was more than 12 times that of activated carbon. Compared with ordinary boron nitride, the adsorption capacity of P-BN was more than four times higher. After this series of experiments, it was concluded that P-BN can be used as an adsorbent of heavy metals in the flue gas. Furthermore, because of the difference in the adsorption selectivity of heavy metals, P-BN can contribute to the separation and recovery of heavy metals.

The thermogravimetry–differential scanning calorimeter (TG-DSC) analysis experiment of P-BN after zinc adsorption was carried out under the nitrogen atmosphere at a heating rate of 15°C min$^{-1}$. As shown in figure 9, first of all, the volatilization of water occurs, mainly before 150°C; then, the zinc chloride adsorbed by P-BN volatilized, which occurs between about 150°C and 950°C; the quality of the sample tends to be stable after 950°C. The experimental results prove that the thermal stability of P-BN under nitrogen atmosphere can be maintained up to 1100°C, which ensures the high-temperature regeneration of P-BN after heavy metals adsorption.

# 4. Conclusion

We successfully synthesized P-BN by using phosphorus as a precursor for boron nitride. The P-BN prepared in this study presented a high specific surface area and a variety of pore structures. The highest specific surface area of the P-BN reached 837.08 m$^2$ g$^{-1}$ at a calcination time of 4 h and doping ratio of 0.50. The experimental research demonstrated that P-BN can adsorb heavy metals from flue gas, and it presents better adsorption capacity for zinc and copper. In particular, P-BN has a stronger adsorption selectivity for zinc than for other heavy metals, and its adsorption capacity for zinc was 5–38 times higher than for other heavy metals. The maximum adsorption capacities at optimum adsorption temperature were 69.45 and 53.80 mg g$^{-1}$ in a single heavy metal atmosphere. Therefore, the obtained P-BN can provide a reference to the development of adsorbents for heavy metals from flue gas and the application of environmental heavy metal pollution control. In future research, we will conduct more in-depth experiments on the removal of heavy metals in the actual flue gas environment in order to analyse the difference of the adsorption mechanism between the actual flue gas and the simulated flue gas. Therefore, we will discuss the feasibility and application of P-BN in the field of heavy metals removal from flue gas.

**Ethics.** This article does not present research with ethical considerations.

**Data accessibility.** Our data are deposited at the Dryad Digital Repository: http://dx.doi.org/10.5061/dryad.k98sf7m3d [37].

**Authors' contributions.** Y.L., H.L. and R.L. designed the study. Y.L. and H.L. prepared the samples for analysis. H.L. and X.S. collected and analysed the data. Y.L., H.L., R.L. and S.S. interpreted the results and wrote the manuscript. All authors gave final approval for publication.

**Competing interests.** The authors declare no competing interests.

**Funding.** Financial support was obtained from the National Natural Science Foundation of China (grant nos. 51706147 and 51576134) and the Educational Commission of Liaoning Province of China (grant no. L201728).

**Acknowledgements.** We would like to thank the support of the specialists Haijun Zhang and Zhenquan Fang of the Key Laboratory of Clean Energy, Shenyang Aerospace University, for their assistance with the BET and ICP

analyses. We also would like to thank Zhenyu Zhai and Yang Chen, who collected relevant data on the adsorption capacity of common boron nitride and activated carbon for heavy metals in flue gas.

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
