## [Reviewer comments · Royal Society Open Science]

Review History

RSOS-200079.R0 (Original submission)

Review form: Reviewer 1

Is the manuscript scientifically sound in its present form?

Yes

Are the interpretations and conclusions justified by the results?

Yes

Is the language acceptable?

Yes

Do you have any ethical concerns with this paper?

No

Have you any concerns about statistical analyses in this paper?

No

Recommendation?

Major revision is needed (please make suggestions in comments)

Comments to the Author(s)

This paper investigates the preparation of phosphorus-doped porous boron nitride and its adsorption of heavy metals from flue gas. The content of this paper is of interest to this journal, thus recommended for publication in Royal Society Open Science. Overall this is well written and thorough. There are a couple of points however that need to be looked at.

1. The entire introduction is very short. Please add some separate paragraphs related to research work that convey a particular aspect of the introduction.
2. Some literature can enrich the content of the introduction part like <https://doi.org/10.1007/s11270-020-04483-4>, Journal of Industrial and Engineering Chemistry 80 (2019) 227–238.
3. Please clearly list and discuss the previous investigations and conclusions on the adsorption of heavy metals performed by the other researchers.
4. The adsorption mechanism of metals should be explained in detail.
5. The SEM image, BET and adsorption and desorption isotherms, some of the changes are not well explained the analysis are not detailed and accurate enough, and the description process is not rigorous enough. Please refer to Materials Chemistry and Physics 233 (2019) 102–112, Environ. Sci.: Water Res. Technol., 2018, 4, 438 for the correction and further analysis.
6. The quality of some Figures is not good. Improved images are needed.
7. Develop more future challenges in the conclusions.

Review form: Reviewer 2

Is the manuscript scientifically sound in its present form?

No

Are the interpretations and conclusions justified by the results?

Yes

Is the language acceptable?

Yes

Do you have any ethical concerns with this paper?

No

Have you any concerns about statistical analyses in this paper?

No

Recommendation?

Major revision is needed (please make suggestions in comments)

Comments to the Author(s)

The reviewed paper describes an interesting experiment of metal sorption from a gas phase at a high temperature. The idea is very interesting and of great practical importance. The work, however, requires additions and clarification of some descriptions.

1. Please present where the test results can be applied. In what (industrial, environmental) conditions are metal emissions at 950 degrees.
2. Methods of metal sorption require clarification:
 - what metal salts were used in the experiments?
 - what is the theoretical volatility of the salts used?
 - what quantities (weight) of metal salts were used?

- how much (weight) sorbent was used?
- how was digestion experiments carried out?

3. Characteristics of sorbents. The authors emphasize the high thermal stability of P-BN in their work. Please present the results of the TG / DTA analysis that will confirm the thermal stability of P-BN. Please, present the results of the XRD analysis of the P-BN sample.

4. The authors, when describing the results of sorption values, use the phrase "strong of adsorption". The strength of metal-binding has not been studied, only the sorption efficiency. "Strong of adsorption" refers to the stability of the metal bond to P-BN, which by the way, should also be studied in the paper. Please, discuss the stability of adsorbed metals at work.

5. Sorption mechanism. On page 8 (lines 31-37, 49-51), the authors mention the mechanism of sorption based on literature studies, but do not present discussions based on the results obtained. I am asking for such a discussion. Certainly, the results of FTIR, SEM, XRD analysis of metal sorption samples would be helpful to carry it out.

Decision letter (RSOS-200079.R0)

26-Mar-2020

Dear Dr Li:

Title: Preparation of phosphorus-doped porous boron nitride and its adsorption of heavy metals from flue gas

Manuscript ID: RSOS-200079

The editor assigned to your manuscript has now received comments from reviewers. We would like you to revise your paper in accordance with the referee and Subject Editor suggestions which can be found below (not including confidential reports to the Editor). Please note this decision does not guarantee eventual acceptance.

Please submit your revised paper before 18-Apr-2020. Please note that the revision deadline will expire at 00.00am on this date. If we do not hear from you within this time then it will be assumed that the paper has been withdrawn. In exceptional circumstances, extensions may be possible if agreed with the Editorial Office in advance. We do not allow multiple rounds of revision so we urge you to make every effort to fully address all of the comments at this stage. If deemed necessary by the Editors, your manuscript will be sent back to one or more of the original reviewers for assessment. If the original reviewers are not available we may invite new reviewers.

Once again, thank you for submitting your manuscript to Royal Society Open Science and I look

forward to receiving your revision. If you have any questions at all, please do not hesitate to get in touch.

On behalf of the Subject Editor Professor Anthony Stace and the Associate Editor Dr Nadia Martinez Villegas.

RSC Associate Editor:

Comments to the Author:

Additional introductory information is needed in order to provide a self stand manuscript for readers to better understand. Detailed experimental descriptions are needed in order to ensure other researchers can reproduce these experiments. Figures and data explanation must be clearly, accurately and thoroughly presented and should be of sufficient quality. A comprehensive comparison with other studies is needed. An accurate use of the word "strength" of adsorption is required as well as the mechanisms of adsorption that is operating in this study.

Reviewers' Comments to Author:

Reviewer: 1

Comments to the Author(s)

This paper investigates the preparation of phosphorus-doped porous boron nitride and its adsorption of heavy metals from flue gas. The content of this paper is of interest to this journal, thus recommended for publication in Royal Society Open Science. Overall this is well written and thorough. There are a couple of points however that need to be looked at.

1. The entire introduction is very short. Please add some separate paragraphs related to research work that convey a particular aspect of the introduction.
2. Some literature can enrich the content of the introduction part like <https://doi.org/10.1007/s11270-020-04483-4>, Journal of Industrial and Engineering Chemistry 80 (2019) 227–238.
3. Please clearly list and discuss the previous investigations and conclusions on the adsorption of heavy metals performed by the other researchers.
4. The adsorption mechanism of metals should be explained in detail.
5. The SEM image, BET and adsorption and desorption isotherms, some of the changes are not well explained the analysis are not detailed and accurate enough, and the description process is not rigorous enough. Please refer to Materials Chemistry and Physics 233 (2019) 102–112, Environ. Sci.: Water Res. Technol., 2018, 4, 438 for the correction and further analysis.
6. The quality of some Figures is not good. Improved images are needed.
7. Develop more future challenges in the conclusions.

Reviewer: 2

Comments to the Author(s)

The reviewed paper describes an interesting experiment of metal sorption from a gas phase at a high temperature. The idea is very interesting and of great practical importance. The work, however, requires additions and clarification of some descriptions.

1. Please present where the test results can be applied. In what (industrial, environmental) conditions are metal emissions at 950 degrees.

2. Methods of metal sorption require clarification:

- what metal salts were used in the experiments?
- what is the theoretical volatility of the salts used?
- what quantities (weight) of metal salts were used?
- how much (weight) sorbent was used?
- how was digestion experiments carried out?

3. Characteristics of sorbents. The authors emphasize the high thermal stability of P-BN in their work. Please present the results of the TG / DTA analysis that will confirm the thermal stability of P-BN. Please, present the results of the XRD analysis of the P-BN sample.

4. The authors, when describing the results of sorption values, use the phrase "strong of adsorption". The strength of metal-binding has not been studied, only the sorption efficiency. "Strong of adsorption" refers to the stability of the metal bond to P-BN, which by the way, should also be studied in the paper. Please, discuss the stability of adsorbed metals at work.

5. Sorption mechanism. On page 8 (lines 31-37, 49-51), the authors mention the mechanism of sorption based on literature studies, but do not present discussions based on the results obtained. I am asking for such a discussion. Certainly, the results of FTIR, SEM, XRD analysis of metal sorption samples would be helpful to carry it out.

Author's Response to Decision Letter for (RSOS-200079.R0)

See Appendix A.

Decision letter (RSOS-200079.R1)

Dear Dr Li:

Title: Preparation of phosphorus-doped porous boron nitride and its adsorption of heavy metals from flue gas

Manuscript ID: RSOS-200079.R1

The editor assigned to your paper has now received comments from reviewers. We would like you to revise your paper in accordance with the referee and Subject Editor suggestions which can be found below (not including confidential reports to the Editor). Please note this decision does not guarantee eventual acceptance.

Please submit a copy of your revised paper before 13-Jun-2020. Please note that the revision deadline will expire at 00.00am on this date. If we do not hear from you within this time then it will be assumed that the paper has been withdrawn. In exceptional circumstances, extensions may be possible if agreed with the Editorial Office in advance. We do not allow multiple rounds of revision so we urge you to make every effort to fully address all of the comments at this stage. If deemed necessary by the Editors, your manuscript will be sent back to one or more of the original reviewers for assessment. If the original reviewers are not available we may invite new reviewers.

On behalf of the Subject Editor Professor Anthony Stace and the Associate Editor Dr Nadia Martinez Villegas.

RSC Associate Editor

Comments to the Author:

Thank you very much for the improvements. The new information results in a more self standing manuscript as well as a more comprehensive comparison of your results with other studies. On the other hand, more detailed experimental descriptions are still needed in order to avoid confusion. Questions arose by Reviewer #2 were answered in the response, but they were not clarified and/or attended in the manuscript. Please be aware that we previously asked you to make the effort to fully address all of the comments as we do not allow multiple rounds of revisions. A thorough revision of the language is also needed, especially for the material recently added. A consistent use of either the term boron nitride or the abbreviation BN along the manuscript will also contribute to the presentation of your draft.

Reviewers' Comments to Author:

Author's Response to Decision Letter for (RSOS-200079.R1)

See Appendix B.

Decision letter (RSOS-200079.R2)

Dear Dr Li:

Title: Preparation of phosphorus-doped boron nitride and its adsorption of heavy metals from flue gas

Manuscript ID: RSOS-200079.R2

It is a pleasure to accept your manuscript in its current form for publication in Royal Society Open Science. The chemistry content of Royal Society Open Science is published in collaboration with the Royal Society of Chemistry.

On behalf of the Subject Editor Professor Anthony Stace and the Associate Editor Dr Nadia Martinez Villegas.

RSC Associate Editor
Comments to the Author:
(There are no comments.)

Reviewer(s)' Comments to Author:

Appendix A

Dear Editor and Reviewers:

Thank you for your letter and for the reviewers' comments concerning our manuscript (RSOS-200079) entitled "Preparation of phosphorus-doped porous boron nitride and its adsorption of heavy metals from flue gas". Those comments are all valuable and very helpful for revising and improving our paper, as well as the important guiding significance to our researches. We have studied comments carefully and have made correction which we hope meet with approval.

Revised portions are marked with blue thoroughly along the whole manuscript. The main corrections in the paper and the responds to the reviewer's comments are as following:

Response to the reviewer#1's comments:

Comment 1:

The entire introduction is very short. Please add some separate paragraphs related to research work that convey a particular aspect of the introduction.

Response: We are sorry for the entire introduction is not enough abundant. We have further enriched the content in the introduction, the specific content are given as follows:

Page 1 lines 33-35: For example, Wang used BN for hydrogen storage experiments and found that the materials exhibit high and reversible H₂ uptake from 1.6 to 2.3 wt% at 77 K and at a relatively low pressure.

Page 1 line 37 - Page 2 line 8: Such as the BN nanomaterial prepared by Singla adsorbed bright green and methyl orange, and explained the feasibility of the interaction between the dye molecules and the adsorbed nanomaterial by density functional theory; Song adsorbed antibiotics through the BN bundles, and found that it has higher adsorption capacity and superior removal percentage than the carbonaceous adsorbent. And his team has demonstrated through repeated experiments that BN bundles have better reusability. Due to the increasing popularity of doping activation theory in materials applications, such as that Shukla introduced carboxylated doping on the membrane material to increase the hydrophilicity of the membrane, thereby significantly improving the water flux and dye removal of the material and Ali uses

silver-doped graphene oxide to enhance the biofouling resistance of the membrane material without affecting the selectivity of the membrane material, so researchers based on density functional theory of boron nitride doping research have gradually become the focus of exploration at present.

Comment 2:

Some literature can enrich the content of the introduction part like <https://doi.org/10.1007/s11270-020-04483-4>, Journal of Industrial and Engineering Chemistry 80 (2019) 227–238.

Response: Your suggestions are very useful for improving our thesis. After consideration we used it to enrich the content of the introduction part. The specific expression is as follows:

Page 2 lines 1-8: Due to the increasing popularity of doping activation theory in materials applications, such as that Shukla introduced carboxylated doping on the membrane material to increase the hydrophilicity of the membrane, thereby significantly improving the water flux and dye removal of the material and Ali uses silver-doped graphene oxide to enhance the biofouling resistance of the membrane material without affecting the selectivity of the membrane material, so researchers based on density functional theory of boron nitride doping research have gradually become the focus of exploration at present.

Comment 3:

Please clearly list and discuss the previous investigations and conclusions on the adsorption of heavy metals performed by the other researchers.

Response: Your suggestions are very useful for improving our thesis. In fact, we have listed and discussed some previous studies and conclusions of other researchers that adsorb heavy metals via BN in the second part of the introduction. For this part, please refer to the Page 2 lines 17-23.

Comment 4:

The adsorption mechanism of metals should be explained in detail.

Response: Your suggestions are very useful for improving our paper. After some consideration, we supplemented the adsorption mechanism. The specific expression is

as follows:

Page 6 lines 4-16: The specific surface area of BN is not less than the specific surface area of activated carbon, which is a typical flue gas adsorbent, and the B-N bonds of BN have a local polarity that C-C bonds do not have and the presence of 28% of ionic bonds. After doping modification, BN enhances the chemical adsorption capacity, and further improves the adsorption capacity of BN on the basis of the original physical adsorption capacity. Lvova discovered the existence of BN vacancies through density functional theory. This phenomenon promotes the chemical activity of boron nitride. The defect regions formed due to the existence of vacancies can provide favorable conditions for the strong covalent bonding between the B and N atoms and the dissociated fragments of the adsorbed molecules. The existence of this vacancy mechanism provides boron nitride with the space and adsorption sites needed for adsorption. And BN has high thermal stability and good chemical stability, so it is an effective material for adsorbing metals from flue gas.

Comment 5:

The SEM image, BET and adsorption and desorption isotherms, some of the changes are not well explained the analysis are not detailed and accurate enough, and the description process is not rigorous enough. Please refer to Materials Chemistry and Physics 233 (2019) 102–112, Environ. Sci.: Water Res. Technol., 2018, 4, 438 for the correction and further analysis.

Response: We are sorry for the some image not well explained and analyzed. In this regard, we conducted a more in-depth analysis of it, and through reading the literatures and further analyzing the images, we reached more abundant conclusions. The specific expression is as follows:

Page 3 lines 21-27: The apparent lamellar structure of the boron nitride fibre is shown in Fig. 1a. It can be clearly seen from the figure that this layered structure can be more clearly reflected in the cross-sectional morphology of the P-BN surface. The rough surface structure of P-BN is the main factor leading to its high specific surface area, as shown in Figure 1b. Such a rough surface structure provides richer adsorption sites and storage space for P introduced by doping, which can contributed to improve its

absorption capacity.

Page 5 lines 12-15: Moreover, the H4-type hysteresis curve is particularly obvious in the adsorption-desorption isotherm, and such characteristic in the adsorption-desorption isotherm indicates that a certain mesoporous structure exists in the P-BN.

Comment 6:

The quality of some Figures is not good. Improved images are needed.

Response: We are very sorry that the quality of some figures is not perfect. After re-exporting the figures in the article, we applied the "wmf" mode, which can be scaled arbitrarily without affecting the image quality to ensure the clarity of the pictures. Hope we can get your approval.

Comment 7:

Develop more future challenges in the conclusions.

Response: We are very grateful for your suggestions, so we discussed in the conclusion more challenges we will face. The specific expression is as follows:

Page 10 lines 6-11: In future research, we will conduct more in-depth experiments on the removal of heavy metals in the actual flue gas environment, and analyze the difference in the adsorption mechanism of the actual flue gas and the simulated flue gas and the difference in the selectivity of heavy metal adsorption. Therefore, we will discuss the feasibility and application of boron nitride in the field of heavy metal flue gas removal.

Response to the reviewer#2's comments:

Comment 1:

Please present where the test results can be applied. In what (industrial, environmental) conditions are metal emissions at 950 degrees.

Response: We are sorry that our description has caused you confusion. In response to your questions, we will make some explanations here. What we call 950°C is the experimental temperature when the metal smoke is generated, and the adsorption temperature is not this temperature. At the present stage, the most widely used method of flue gas removal in the flue gas adsorption process is the use of activated carbon

adsorption at the end of the flue. The adsorption temperature at this time is about 100°C, which is caused by the thermal stability of the activated carbon itself. The boron nitride material we use is a new type of non-carbon based sorbents. It has been found in previous studies that it has high thermal stability and can be stable at 900 ~ 1000°C in the oxygen environment. In practical applications, BN can be adsorbed and removed at the front end of flue gas adsorption to remove heavy metals. The application temperature at this time is about 400 ~ 600°C. Such adsorption removal can also reduce the formation of dioxins in the flue gas, because the heavy metal Cu is a catalyst for the formation of dioxins at high temperatures.

According to the following articles:

- [1] Sushobhan J., David E., Ralph K., et al. Boron Nitride on Cu (111): An Electronically Corrugated Monolayer [J]. Nano Letters, 2012, 12(11).
- [2] REN J., ZHOU J., LUO Z., et al. An Experimental Study on Activated Carbon Sorbents for Gas-Phase Mercury Removal from Flue Gas [J]. Proceedings of the CSEE, 2004, 024(002):171-175.
- [3] REN J., CHEN J., LUO Y., et al. Characteristics of the Mercury Vapor Removal From Flue Gas by Activated Carbon Fibers [J]. Proceedings of the CSEE, 2010, 030(005):28-34.
- [4] Loh K., Mikka N., Sakaguchi I., et al. Thermal stability of the negative electron affinity condition on cubic boron nitride[J]. Applied Physics Letters, 72(23):3023.

Comment 2:

Methods of metal sorption require clarification:

- what metal salts were used in the experiments?
- what is the theoretical volatility of the salts used?
- what quantities (weight) of metal salts were used?
- how much (weight) sorbent was used?
- how was digestion experiments carried out?

Response: We are sorry for the omission of the related introduction of metal salts, and we will give some instructions on this. The metal salts we used in this experiment are easily available halogen metal salts such as FeCl₃ and CuCl₂. After the experimental temperature is reached in the experiment, the metal salt can be converted well to reach the smoke composition required for the experiment. In the course of the experiment,

the basis weight of each metal salt was 5 g, and the basis weight of the adsorbent was 1.5 g. For digestion experiments, we used "Microwave-assisted acid digestion" ("HF-HNO₃-HClO₄" microwave digestion) to perform microwave digestion experiments. The obtained digestion solution was diluted and the heavy metal content was detected by ICP to obtain our experimental data.

Comment 3:

Characteristics of sorbents. The authors emphasize the high thermal stability of P-BN in their work. Please present the results of the TG / DTA analysis that will confirm the thermal stability of P-BN. Please, present the results of the XRD analysis of the P-BN sample.

Response: Thank you for your interest in the thermal stability of P-BN. P-BN is prepared by doping, the main body is still BN material, so it has good thermal stability like BN. Through literature we can clarify its excellent performance, so P-BN still has good thermal stability.

According to the following articles:

- [1] Kostoglou N , Polychronopoulou K , Rebholz C . Thermal and chemical stability of hexagonal boron nitride (h-BN) nanoplatelets[J]. Vacuum, 2015, 112:42-45.
- [2] Loh K., Mikka N., Sakaguchi I., et al. Thermal stability of the negative electron affinity condition on cubic boron nitride[J]. Applied Physics Letters, 72(23):3023.
- [3] Dibandjo P , Bois L , Chassagneux F , et al. Thermal stability of mesoporous boron nitride templated with a cationic surfactant[J]. journal of the european ceramic society, 2007, 27(1):313-317.
- [4] Han M , Yu J . Pressure-induced vapor synthesis, formation mechanism, and thermal stability of well-dispersed boron nitride spheres [J]. Diamond & Related Materials, 2018 (87): 10-17.

Comment 4:

The authors, when describing the results of sorption values, use the phrase "strong of adsorption". The strength of metal-binding has not been studied, only the sorption efficiency. "Strong of adsorption" refers to the stability of the metal bond to P-BN, which by the way, should also be studied in the paper. Please, discuss the stability of

adsorbed metals at work.

Response: Your comments are very useful for our paper, so that we understand the evaluation methods of adsorption stability and the meaning of expression applications. We did not find the "strong of adsorption" you mentioned in the article. After discussion, we think that you may be confused about the use of "stronger adsorption", so we will explain it here. "Stronger adsorption" as used in this article only means that P-BN has better adsorption capacity for Zn and Cu or P-BN has better adsorption capacity for heavy metals compared to activated carbon and ordinary BN. Therefore, we hope our response will satisfy you.

Comment 5:

Sorption mechanism. On page 8 (lines 31-37, 49-51), the authors mention the mechanism of sorption based on literature studies, but do not present discussions based on the results obtained. I am asking for such a discussion. Certainly, the results of FTIR, SEM, XRD analysis of metal sorption samples would be helpful to carry it out.

Response: As you said, we have introduced the research conclusions of related literature in the article to verify our experimental conclusions. But such references are used to prove the correctness of the conclusions derived from our experimental data. These related data discussions are reflected to a certain extent in the article, which may be because it appeared before the relevant cited literature and did not attract your attention. I apologize for this. The experimental data in this article is based on the detection results of ICP to discuss the adsorption capacity of P-BN, so you may have some doubts. We hope that our discussion method will be approved by you. The related discussion can be found at Page 7 line 24 - Page 8 line 4 and Page 8 line 13-18, we hope to get your approval.

We tried our best to improve the manuscript and made some changes in the manuscript. These changes will not influence the content and framework of the paper. We appreciate for Editors and Reviewers' warm work earnestly, and hope that the correction will meet with approval.

Once again, thank you very much for your comments and suggestions.

Yours sincerely,

Yanlong Li, Hongxi Li, Rundong Li*, Xin Su, Shengqiang Shen

College of Energy and Environment, Shenyang Aerospace University, Key Laboratory
of Clean Energy, Liaoning Province

Appendix B

Dear Reviewers:

Thank you for your letter and for the comments concerning our manuscript (RSOS-200079.R1) entitled “Preparation of phosphorus-doped boron nitride and its adsorption of heavy metals from flue gas”. Those comments are all valuable and very helpful for revising and improving our paper, as well as the important guiding significance to our researches. We have studied comments carefully and have made correction which we hope meet with approval.

Revised portions are marked with blue thoroughly along the whole manuscript.

Regarding the experimental scheme, we have explained in detail in the ‘Experiment’ part of the article, we added some experimental descriptions, just like ‘what metal salts were used in the experiments?’ and ‘what quantities (weight) of metal salts were used?’ which is the questions arose by Reviewer #2. We hope that these descriptions can avoid possible confusion issues. We have revised the article many times, unifying the terms used in the article and the abbreviations of phosphorus-doped boron nitride to make the expression more accurate.

In addition, we revised and supplemented the manuscript based on some comments arose by Reviewer #2 in the previous review.

Comment 1:

Please present where the test results can be applied. In what (industrial, environmental) conditions are metal emissions at 950 degrees.

Response: We are sorry that our description has caused you confusion. In response to your questions, we will make some explanations here. What we call 950°C is the experimental temperature when the metal smoke is generated, and the adsorption temperature is not this temperature. At the present stage, the most widely used method of flue gas removal in the flue gas adsorption process is the use of activated carbon adsorption at the end of the flue. The adsorption temperature at this time is about 100°C, which is caused by the thermal stability of the activated carbon itself. The boron nitride material we use is a new type of non-carbon based sorbents. It has been found in previous studies that it has high thermal stability and can be stable at 900 ~ 1000°C in

the oxygen environment. In practical applications, BN can be adsorbed and removed at the front end of flue gas adsorption to remove heavy metals. The application temperature at this time is about 400 ~ 600°C. Such adsorption removal can also reduce the formation of dioxins in the flue gas, because the heavy metal Cu is a catalyst for the formation of dioxins at high temperatures.

According to the following articles:

- [1] Sushobhan J., David E., Ralph K., et al. Boron Nitride on Cu (111): An Electronically Corrugated Monolayer [J]. Nano Letters, 2012, 12(11).
- [2] REN J., ZHOU J., LUO Z., et al. An Experimental Study on Activated Carbon Sorbents for Gas-Phase Mercury Removal from Flue Gas [J]. Proceedings of the CSEE, 2004, 024(002):171-175.
- [3] REN J., CHEN J., LUO Y., et al. Characteristics of the Mercury Vapor Removal From Flue Gas by Activated Carbon Fibers [J]. Proceedings of the CSEE, 2010, 030(005):28-34.
- [4] Loh K., Mikka N., Sakaguchi I., et al. Thermal stability of the negative electron affinity condition on cubic boron nitride[J]. Applied Physics Letters, 72(23):3023.

Comment 2:

Methods of metal sorption require clarification:

- what metal salts were used in the experiments?
- what is the theoretical volatility of the salts used?
- what quantities (weight) of metal salts were used?
- how much (weight) sorbent was used?
- how was digestion experiments carried out?

Response: We are sorry for the omission of the related introduction of metal salts, and we will give some instructions on this. The metal salts we used in this experiment are easily available halogen metal salts such as $ZnCl_2$ and $CuCl_2$. After the experimental temperature is reached in the experiment, the metal salt can be converted well to reach the smoke composition required for the experiment. In the course of the experiment, the basis weight of each metal salt was 5 g, and the basis weight of the adsorbent was 1.5 g. For digestion experiments, we used "Microwave-assisted acid digestion" ("HF-HNO₃-HClO₄" microwave digestion) to perform microwave digestion experiments. The obtained digestion solution was diluted and the heavy metal content was detected

by ICP to obtain our experimental data.

We are sorry for the introduction is not enough abundant. We have further enriched the experimental program, the specific content are given as follows:

Page 3 lines 1-8: A sufficient amount of heavy metal chlorides were used as the source of heavy metals, was placed in the constant temperature zone of the tube furnace, and the P-BN was placed in the heavy metal adsorption zone to absorb the heavy metals from the flue gas. In the course of the experiment, the basis weight of each heavy metal chloride was 5g, and the basis weight of the adsorbent was 1.5g. The tube furnace was heated to 950°C to ensure the occurrence of heavy metal fumes. The adsorbed P-BN was subjected to digestion experiments, using microwave-assisted acid digestion (HF-HNO₃-HClO₄). Then, the concentration of heavy metals in the digestion solution were analysed by ICP.

Comment 3:

Characteristics of sorbents. The authors emphasize the high thermal stability of P-BN in their work. Please present the results of the TG / DTA analysis that will confirm the thermal stability of P-BN. Please, present the results of the XRD analysis of the P-BN sample.

Response: Thank you for your interest in the thermal stability of P-BN. P-BN is prepared by doping, the main body is still BN material, so it has good thermal stability like BN. Through literature we can clarify its excellent performance, so P-BN still has good thermal stability.

According to the following articles:

- [1] Kostoglou N , Polychronopoulou K , Rebholz C . Thermal and chemical stability of hexagonal boron nitride (h-BN) nanoplatelets[J]. Vacuum, 2015, 112:42-45.
- [2] Loh K., Mikka N., Sakaguchi I., et al. Thermal stability of the negative electron affinity condition on cubic boron nitride[J]. Applied Physics Letters, 72(23):3023.
- [3] Dibandjo P , Bois L , Chassagneux F , et al. Thermal stability of mesoporous boron nitride templated with a cationic surfactant[J]. journal of the european ceramic society, 2007, 27(1):313-317.
- [4] Han M , Yu J . Pressure-induced vapor synthesis, formation mechanism, and thermal stability of well-dispersed boron nitride spheres [J]. Diamond & Related Materials, 2018 (87): 10-17.

According to your suggestion, we analyzed the adsorbed samples by TG-DSC, the

specific content are given as follows:

Page 9 line 16- 22: The TG-DSC analysis experiment of P-BN after zinc adsorption was carried out under the nitrogen atmosphere at a heating rate of 15°C/min. As shown in Fig. 9, first of all, the volatilization of water occurs, mainly before 150°C; then, the zinc chloride adsorbed by P-BN volatilized, which occurs between about 150°C and 950°C; the quality of the sample tends to be stable after 950°C. The experimental results prove that the thermal stability of P-BN under nitrogen atmosphere can be maintained up to 1100°C, which ensures the high-temperature regeneration of P-BN after heavy metals adsorption.

Fig.9 TG-DSC curve of P-BN after zinc adsorption (15°C/min, N₂ atmosphere)

Comment 4:

The authors, when describing the results of sorption values, use the phrase "strong of adsorption". The strength of metal-binding has not been studied, only the sorption efficiency. "Strong of adsorption" refers to the stability of the metal bond to P-BN, which by the way, should also be studied in the paper. Please, discuss the stability of adsorbed metals at work.

Response: Your comments are very useful for our paper, so that we understand the evaluation methods of adsorption stability and the meaning of expression applications. We did not find the "strong of adsorption" you mentioned in the article. After discussion, we think that you may be confused about the use of "stronger adsorption", so we will explain it here. "Stronger adsorption" as used in this article only means that P-BN has better adsorption capacity for Zn and Cu or P-BN has better adsorption capacity for heavy metals compared to activated carbon and ordinary BN. Therefore, we hope our response will satisfy you.

Comment 5:

Sorption mechanism. On page 8 (lines 31-37, 49-51), the authors mention the mechanism of sorption based on literature studies, but do not present discussions based on the results obtained. I am asking for such a discussion. Certainly, the results of FTIR, SEM, XRD analysis of metal sorption samples would be helpful to carry it out.

Response: As you said, we have introduced the research conclusions of related literature in the article to verify our experimental conclusions. But such references are used to prove the correctness of the conclusions derived from our experimental data. These related data discussions are reflected to a certain extent in the article, which may be because it appeared before the relevant cited literature and did not attract your attention. I apologize for this. The experimental data in this article is based on the detection results of ICP to discuss the adsorption capacity of P-BN, so you may have some doubts. We hope that our discussion method will be approved by you. The related discussion can be found at Page 7 line 24 - Page 8 line 4 and Page 8 line 13-18, we hope to get your approval.

We tried our best to improve the manuscript and made some changes in the manuscript. These changes will not influence the content and framework of the paper. We appreciate for Editors and Reviewers' warm work earnestly, and hope that the correction will meet with approval.

Once again, thank you very much for your comments and suggestions.

Yours sincerely,

Yanlong Li, Hongxi Li, Rundong Li*, Xin Su, Shengqiang Shen

College of Energy and Environment, Shenyang Aerospace University, Key Laboratory of Clean Energy, Liaoning Province